# CX3CR1-Targeted PLGA Nanoparticles Reduce Microglia Activation and Pain Behavior in Rats with Spinal Nerve Ligation

**DOI:** 10.3390/ijms21103469

**Published:** 2020-05-14

**Authors:** Chan Noh, Hyo Jung Shin, Seounghun Lee, Song I Kim, Yoon-Hee Kim, Won Hyung Lee, Dong Woon Kim, Sun Yeul Lee, Young Kwon Ko

**Affiliations:** 1Department of Anesthesiology and Pain Medicine, Chungnam National University Hospital, 282 Munhwa-ro, Jung-gu, Daejeon 35015, Korea; jyfchrh@gmail.com (C.N.); anelee1982@cnuh.co.kr (S.L.); yhkim040404@gmail.com (Y.-H.K.); whlee@cnu.ac.kr (W.H.L.); 2Department of Medical Science, 3 Department of Anatomy and Cell Biology, 4 Brain Research Institute, School of medicine, Chungnam National University, Daejeon 35015, Korea; shinhyo1013@gmail.com (H.J.S.); kthddl2295@gmail.com (S.I.K.); visnu528@cnu.ac.kr (D.W.K.)

**Keywords:** microglia, CX3CR1, Poly(D,L-lactic-co-glycolic acid) (PLGA) nanoparticles, siRNA, neuropathic pain, spinal nerve ligation

## Abstract

Activation of CX3CR1 in microglia plays an important role in the development of neuropathic pain. Here, we investigated whether neuropathic pain could be attenuated in spinal nerve ligation (SNL)-induced rats by reducing microglial activation through the use of poly(D,L-lactic-co-glycolic acid) (PLGA)-encapsulated CX3CR1 small-interfering RNA (siRNA) nanoparticles. After confirming the efficacy and specificity of CX3CR1 siRNA, as evidenced by its anti-inflammatory effects in lipopolysaccharide-stimulated BV2 cells in vitro, PLGA-encapsulated CX3CR1 siRNA nanoparticles were synthesized by sonication using the conventional double emulsion (W/O/W) method and administered intrathecally into SNL rats. CX3CR1 siRNA-treated rats exhibited significant reductions in the activation of microglia in the spinal dorsal horn and a downregulation of proinflammatory mediators, as well as a significant attenuation of mechanical allodynia. These data indicate that the PLGA-encapsulated CX3CR1 siRNA nanoparticles effectively reduce neuropathic pain in SNL-induced rats by reducing microglial activity and the expression of proinflammatory mediators. Therefore, we believe that PLGA-encapsulated CX3CR1 siRNA nanoparticles represent a valuable new treatment option for neuropathic pain.

## 1. Introduction

Neuropathic pain is caused by various lesions or diseases of the somatosensory system [1]. The associated neuropathic symptoms are often influenced by external factors, such as the weather, as well as the condition of the patient, and may be accompanied by comorbidities such as allodynia and hyperalgesia that can make daily life difficult. In addition, patients often suffer from a vicious cycle of depression and insomnia driven by their underlying neuropathic condition. Successful treatment of neuropathic pain is therefore of critical importance in the area of pain management. A wide variety of treatments such as nerve blockage and pharmacological remedies may be prescribed for patients with neuropathic pain, though the efficacy of these treatments is limited due to a lack of understanding of the mechanisms underlying neuropathic pain maintenance and development [2].

In general, when a nerve is damaged, neurotransmitters such as glutamate, ATP, and substance P are released form presynaptic terminals of the primary afferent neurons in the dorsal root ganglia [3]. Release of these neurotransmitters causes nociceptive pain by depolarizing postsynaptic neurons in the spinal cord [4]. These molecules subsequently bind to the corresponding receptors of the glia and microglia of the dorsal horn of the spinal cord and activate the microglia [5], leading to the release of various cytokines and other diffusible molecules in the activated microglia, exacerbating the symptoms of the disease [6]. 

The correlation between neuropathic pain and microglia is well established, with strong evidence of both morphological and pathological mechanisms of disease [7]. In the spinal nerve ligation (SNL) rat model, the microglia in the dorsal horn of the spinal cord showed significant changes in morphology and in the number of spots staining with specific microglial markers [8,9,10,11].

CX3CL1 is a neurotransmitter with a unique structure belonging to the chemokine family, with signaling mediated via ligation to its cognate receptor, CX3CR1. Unlike other chemokines, CX3CL1 is primarily expressed in the kidney, heart, lung, and brain, with no expression seen in leukocytes [12,13]. Many studies have shown that chemokines play an important role in the progression of pain [14]. Intrathecal injection of CX3CL1 in naive rats was shown to induce thermal hyperalgesia and mechanical allodynia, whereas injection of anti-CX3CR1 antibodies reduced the incidence of allodynia and hyperalgesia [15]. Furthermore, results of the SNL model revealed increased expression of CX3CL1 in astrocytes and upregulation of CX3CR1 in microglia of the spinal cord. These results confirm that CX3CL1 and its receptor, CX3CR1, represent promising new targets for the treatment of neuropathic pain [16].

Recently, gene therapies based on small interfering RNAs (siRNA) that silence disease-related genes have attracted considerable attention due to their potential to treat genetic disorders and other intractable conditions [17]. siRNA has been shown to interfere with the expression of specific protein coding genes through the formation of an RNA-induced silencing complex, resulting in the degradation of mRNA molecules. siRNAs may therefore be a good option for the treatment of neuropathic pain. However, a major limitation on the use of siRNA, both in vitro and in vivo, is the potential for off-target effects mediated by the strong anion charge of naked siRNAs and the high degradation of nuclease-sensitive siRNA. Thus, the success of siRNA applications depends on the preparation of a vector suitable for delivering a therapeutic gene to a specific target [18].

Poly(D,L-lactic-co-glycolic acid) (PLGA) is a Food and Drug Administration (FDA)-approved method for delivering siRNA to specific targets [19]. PLGA has already proven effective as a vehicle for the treatment of pain and for the delivery of certain drugs [20]. In particular, PLGA has numerous features that are highly amenable to siRNA delivery. First, they are small enough for efficient tissue infiltration and cellular uptake. Second, the siRNA is readily encapsulated within the PLGA matrix, providing physical protection against RNase activity. Third, PLGA particles have highly controllable and modifiable degradation profiles, enabling targeted drug release depending on the weight of the target molecule and the structure and composition of the nanoparticles [21].

In this study, we examined the use of PLGA-encapsulated CX3CR1 siRNA nanoparticles for the treatment of neuropathic pain in an SNL model. To test our hypothesis, we investigated whether PLGA-encapsulated CX3CR1 siRNA nanoparticles can reduce pain behavior in SNL-induced neuropathic pain rats.

## 2. Results

### 2.1. CX3CR1 siRNA Blocks CX3CR1 Expression in BV2 Cells and Reduces Expression of Pro-Inflammatory Mediators in Lipopolysaccharide-Stimulated BV2 Cells

Before generating PLGA-encapsulated CX3CR1 siRNA nanoparticles, we confirmed the efficacy and specificity of CX3CR1 siRNA in microglial cells. First, to evaluate whether CX3CR1 siRNA can effectively reduce the CX3CR1 protein level in microglial cells, the experimental group was treated with BV2 cells with CX3CR1 siRNA for two days, while the control group used scrambled (sc) siRNA. CX3CR1 protein levels were then detected by Western blot, and the CX3CR1/ACTB ratio was quantified. The expression of CX3CR1 protein was reduced by approximately 50% in CX3CR1 siRNA-treated rats compared to the control group (Figure 1A,B).

Lipopolysaccharide (LPS) activates various cell types, including microglial cells, resulting in the transcription of a wide range of proinflammatory mediators [22]. Based on these observations, we investigated whether CX3CR1 siRNA treatment of microglial cells activated by LPS could downregulate mRNA expression of proinflammatory genes such as TNF-α, iNOS, and COX-2. mRNA expression of proinflammatory-related genes in the scrambled control siRNA (sc siRNA)- and LPS-treated BV2 cells increased by 5- and 30-fold respectively, compared to sc siRNA-treated BV2 cells (Figure 2C). In contrast, CX3CR1 siRNA treatment significantly reduced gene expression by LPS compared to both sc siRNA- and LPS-treated BV2 cells (Figure 1C). Together, these results show that CX3CR1 siRNA significantly reduced protein expression in microglial cells, resulting in a decrease in proinflammatory mediators in microglial cells activated by LPS. 

### 2.2. Preparation and Characterization of PLGA-Encapsulated CX3CR1 siRNA Nanoparticles 

Several factors were considered when selecting a gene delivery system to effectively deliver CX3CR1 siRNA to microglia in the spinal cord, including efficacy, cost, safety, and convenience. In this study, we used PLGA, a substance that has already have been proven biodegradable and biocompatible in humans by the US Food and Drug Administration (FDA) [19]. PLGA nanoparticles were prepared through sonication using the conventional double emulsion (W/O/W) method, as reported in our previous papers [23,24,25], and hydrophilic siRNA was effectively encapsulated in PLGA nanoparticles (Figure 2A). The uniformity and morphology of the nanoparticles were confirmed by scanning electron microscopy (SEM) (Figure 2B). In addition, Zetasizer measurements revealed the formation of monodisperse particles (PDI ≤ 0.2) within the desired size range for siRNA encapsulation. The average size and zeta potential of PLGA-encapsulated CX3CR1 siRNA nanoparticles measured using the Zetasizer ZS90 were 227.8 nm and −34.3 mV, respectively (Figure 2C,D).

### 2.3. Mechanical Allodynia and Upregulated Microglia Activation in SNL-Induced Rats

To evaluate pain behavior caused by neuropathic pain, we used SNL, a well-established model for evaluating neuropathic pain in rats [26]. Before SNL surgery, rats were subjected to the von Frey filament test, and only rats that passed the predefined baseline (≥10 g) were used for subsequent analyses to avoid affecting the results of the behavior test. Mechanical allodynia in the rats was evaluated at 3, 5, 7, 10, and 14 days after surgery. For all rats undergoing SNL surgery, the mechanical threshold for the ipsilateral-side paws began to decrease on day 3, peaked on day 10, and persisted until day 14 (Figure 3A). In contrast, the sham group did not show mechanical allodynia on the ipsilateral-side paws (Figure 3B). 

After SNL surgery, microglia are activated in the ipsilateral dorsal horn of the spinal cord. Activation of microglia on the contralateral side was not observed. To assess the activation of microglia, tissue sections were immuno-stained using an anti-Iba 1 antibody. In the sham group, microglia were not activated in the dorsal horn on either side of the spinal cord; however, in the SNL group, the microglia on the ipsilateral side was dramatically activated in the dorsal horn as well as in the ventral horn compared to the contralateral side at 3 days post-operation (Figure 3C,D).

### 2.4. Intrathecal Injection of PLGA-Encapsulated CX3CR1 siRNA Nanoparticles Reduces Mechanical Allodynia in SNL-Induced Rats

To investigate whether PLGA-encapsulated CX3CR1 siRNA nanoparticles can alleviate pain behavior caused by neuropathic pain, sc siRNA or PLGA-encapsulated CX3CR1 siRNA nanoparticles were injected into the intervertebral space of SNL-induced rats on the fourth day, when the pain was stabilized after surgery, using a Hamilton syringe (Figure 4A). The mechanical threshold was measured using von Frey filaments at 3, 5, 7, 9, 10, 12, and 14 days post-surgery. In the CX3CR1 siRNA-treated group, mechanical allodynia was improved at day 6 after injection, with the most significant effects seen on day 10 (Figure 4B). In contrast, no change was observed in the group injected with sc siRNA-encapsulated PLGA nanoparticles. 

To confirm whether PLGA-encapsulated CX3CR1 siRNA nanoparticles reduced the expression of CX3CR1 in the spinal cord, L5 spinal dorsal horn tissues were collected at 6 days post-intrathecal injection that had the best pain relief effect. CX3CR1 siRNA treatment significantly reduced the expression of CX3CR1 protein, as determined by Western blot using anti-CX3CR1 antibodies (Figure 4C,D). Together, these data show that intrathecal injection of PLGA-encapsulated CX3CR1 siRNA nanoparticles significantly improved SNL-induced pain behavior. 

### 2.5. PLGA-Encapsulated CX3CR1 siRNA Nanoparticles Reduce Microglial Activation and Downregulate the Expression of Neuropathic Pain-Related Genes in the Spinal Dorsal Horn of the SNL-Induced Rat Spinal Cord

Increased expression of CX3CR1 in microglia plays an important role in neuropathic pain. Thus, we hypothesized that inhibiting the expression of CX3CR1 using PLGA-encapsulated CX3CR1 siRNA nanoparticles could reduce microglia activation and alleviate neuropathic pain. As expected, microglial activation was significantly reduced, in terms of both the absolute number and cell density, in the ipsilateral dorsal horn on day 3 post-injection in the CX3CR1 siRNA group compared to the sc siRNA group (Figure 5A,B).

Next, we examined changes in neuropathic pain-related inflammatory genes’ expression. Spinal dorsal horn tissues on the ipsilateral side were collected separately on day 3 post-injection, and the mRNA levels of proinflammatory genes including TNF-α, IL-1β, IL-6, and COX-2 were measured by qRT-PCR. mRNA levels of these proinflammatory genes were significantly decreased in the group injected with PLGA-encapsulated CX3CR1 siRNA nanoparticles compared to the group injected with sc siRNA-encapsulated PLGA nanoparticles (Figure 4E).

Together, these data indicate that PLGA-encapsulated CX3CR1 siRNA nanoparticles inhibit the expression of CX3CR1, thereby inhibiting microglial activation and the expression of proinflammatory genes and alleviating neuropathic pain in SNL-induced rats. 

## 3. Discussion

Neuropathic pain is a chronic condition that can be difficult to treat. Various methods have been developed to treat neuropathic pain, including drug therapies, such as analgesics, antiepileptic drugs, tricyclic antidepressants (TCA), serotonin-norepinephrine reuptake inhibitors (SNRIs), and opioids, as well as mechanical interventions such as nerve blockage, spinal cord stimulation, and intrathecal pumps; however, the effectiveness of these treatments remains unclear [27]. The development of new methods for the treatment of neuropathic pain are therefore necessary. Here, we investigated the use of siRNA-encapsulated nanoparticles targeting microglial-specific genes to reduce microglia activation in neuropathic pain. 

Activated microglia in the spinal cord play an important role in the development of neuropathic pain. Activated microglia have been shown to induce and sustain neuropathic pain by stimulating secretion of various proinflammatory factors, such as TNF-α, IL-1β, and IL-6. Therefore, reduction of microglial activation represents a promising therapeutic target for the reduction of neuropathic pain. Drugs that reduce the activity of microglia include minocycline and pentoxifylline. While effective, these drugs have the potential to generate cytotoxicity when used at high doses [28] and have limited use due to their short half-life [29]. Accordingly, gene therapies that can reduce the activation of microglia and can be used for longer durations are of particular interest. 

CX3CR1 is upregulated in spinal microglia upon nerve injury and is involved in the development and maintenance of neuropathic pain [30]. In addition, upregulation of CX3CR1 is essential for activation of the p38 MARK pathway, which contributes to the development of neuropathic pain [2]. Therefore, we attempted to reduce neuropathic pain in rats with SNL-induced neuropathic pain by reducing the activation of microglia through silencing of the CX3CR1 gene expression in the spinal microglia. First, the anti-inflammatory effect of CX3CR1 siRNA on LPS-stimulated BV2 cells in microglial cells was confirmed. Subsequently, synthesized PLGA-encapsulated CX3CR1 siRNA nanoparticles were injected intrathecally to evaluate the effect on neuropathic pain via intrathecal injection of PLGA-encapsulated CX3CR1 siRNA nanoparticles. As a result, pain behavior was clearly alleviated, and microglia activation was significantly suppressed in SNL-induced neuropathic pain rats injected with PLGA-encapsulated CX3CR1 siRNA nanoparticles. Moreover, the synthesis of proinflammatory mediators such as TNF-α, IL-1β, and COX2 was inhibited by PLGA-encapsulated CX3CR1 siRNA nanoparticles. These results suggest that the silence of the CX3CR1 gene inhibits the activation of microglia, thereby alleviating neuropathic pain. However, even on day 7 after the intrathecal injection, which showed a greater effect, these rats continued to show lower mechanical thresholds than did sham rats and contralateral controls. Thus, it is not possible to control all neuropathic pain by regulating only the activity of microglia, though significant symptom relief can be expected. 

In this study, nanoparticles were used to effectively deliver CX3CR1 siRNA. Nanoparticles are a suitable system for most drug administration routes. Various natural and synthetic polymers have been studied for several years to improve the biocompatibility and biodegradability of nanoparticles. Among them, PLGA used in this study is the most commonly used polymer because of its biocompatibility and resorption through natural pathways. As such, research on nanoparticles’ manufacturing continues. As a result, nanoparticles are attractive mediators for gene silencing and are an effective means of treating neuropathic pain due to their various advantages, including efficient cell uptake, sustained intracellular drug release, favorable safety profiles, low cost, and ease of manufacture. However, the disadvantage of PLGA nanoparticles is that siRNA encapsulation efficiency and delivery power are lower than that of recombinant viruses. Therefore, further research will be needed.

## 4. Materials and Methods 

### 4.1. Animals and Materials

Sprague-Dawley rats (6-week-old males, 150~200 g) were purchased form Daehan Bio Link (DBL, Chung-buk, Republic of Korea) and placed in the study habitat 1 week before the experiment to allow for adaptation to the environment. Rats were housed three per cage in a controlled environment with a 12 h light/dark cycle and free access to food and water. This study was approved by the Animal Care and Use Committee at the Chungnam National University (CNUH-019-A0040) and followed the ethical guidelines of the National Institutes of Health and the International Association for the Study of Pain [31]. The number of animals used in the experiment was 29 per each group, and only rats that passed the basic baseline were used. siRNAs in this study were purchased from Thermo Fisher Scientific (MO, USA) unless otherwise noted. Scrambled siRNA (cat 12935400) and CX3CR1 siRNA (cat 1330001) for the preparation of PLGA nanoparticles were obtained from Thermo Fisher Scientific.

### 4.2. siRNA Transfection into BV2 Cells

BV2 cells were grown in Dulbecco’s modified Eagle’s medium (DMEM) with 10% fetal bovine serum (FBS). All cultures were kept at 37 °C in a 5% CO_2_ humidified incubator. Scrambled siRNA or CX3CR1 siRNA was introduced into BV2 cells using the Lipofectamine 2000 transfection reagent (Thermo Fisher Scientific).

### 4.3. Western Blot and Immunohistochemistry

Spinal cords collected for the purpose of immunostaining were cryopreserved at –80 °C after fixing in 4% PFA and cryoprotected in 30% sucrose for 2–3 days. Thereafter, spinal cords were sectioned coronally at 35 µm using a freezing microtome, and sections were stored in PBS. Nuclei were counterstained with Hoechst 33342 and visualized with a confocal microscope. The immune densities in the graphs were quantified using Image J software.

### 4.4. Quantitative RT-PCR

Total RNA from BV2 cells was isolated using TRIzol reagent according to the manufacturer’s protocol (Gene All, RoboExTM). RNA concentrations were quantified using a Nanodrop spectrometer (Thermo scientific). cDNA was prepared from total RNA using a commercially available kit (Enzynomics, B201). Quantitative polymerase chain reaction (qPCR) was performed using the AriaMax Real-time PCR system (Agilent Technologies) with the TOPreal qPCR 2X premix (SYBR Green with low ROX). The primers used for rat TNF-α, IL-1β, and COX2 were as follows: TNF-α 5′-AGATGTGGAACTGGCAGAGG-3′, and antisense 5′-CCCATTTGGGAACTTCTCCT-3′; IL-1β, 5′-CAGCAGCATCTCGACAAGAG-3′, and antisense 5′-CATCATCCCACGAGTCACAG-3′ COX2, 5′-CAGTATCAGAACCGCATTGCC-3′, and antisense 5′-GAGCAAGTCCGTGTTCAAGGA-3′.

### 4.5. PLGA Nanoparticle Preparation 

PLGA nanoparticles from the Nanoglia Company (Daejeon, Republic of Korea) were prepared as reported previously, with minor modifications [23]. Briefly, 200 µL of 20 µM siRNA in DEPC water (Enzynomics) was added on a drop-by-drop basis to 800 µL of dichloromethane (DCM) containing 25 mg of PLGA (Corbion, Amsterdam, The Netherlands), emulsified by sonication (10%, 30 s; Vibra-CellTM VCX 130; Sonics, Newtown, CT, USA) into a primary W1/O emulsion, and then further emulsified by sonication to form a W1/O/W2 double emulsion. The resulting product was then diluted with 5 mL of 1% PVA1500 and incubated on a magnetic stir plate for 3 h at room temperature to evaporate the DCM. The PLGA nanoparticles were collected by ultracentrifugation (Optima Max Ultracentrifuge, Beckman Coulter, CA, USA) at 38,000 *g* for 10 min at 4 °C, washed once with deionized RNAase-free water, resuspended in water, and freeze-dried. The physical characteristics were then analyzed with the Zetasizer Nano ZS (Malvern Instruments, Malvern, UK) and electron microscopy.

### 4.6. Induction of Neuropathic Pain by L5 SNL

Neuropathic pain in rats was achieved through the widely used L5 nerve root spinal nerve ligation (SNL) method [4,26]. Rats were anesthetized in a prone position by intramuscular injection of 8 mg of Alfaxan (Jurox Animal Health, North Kansas City, USA) into the right hind limb. After applying the potadine ball on the lower back, the paraspinal muscle on the left side was separated using a 2 cm skin incision. The L5 transverse process was then removed to expose the L4 and L5 vertebrae, and the L5 nerve was carefully isolated from the L4 nerve without damage. The isolated L5 nerve was ligated three times using 3-0 silk thread (Ethicon, Diegem, Belgium). The wound was sutured after confirming complete hemostasis. In the sham group, the L5 vertebrae was exposed in the same way as in the SNL group, but the L5 nerve ligation was not performed. 

### 4.7. Pain Behavior Test with Von Frey Filaments

The pain behavior test was evaluated using manual von Frey filaments (NC12775-99, North Coast Medical and Rehabilitation Products, CA, USA), as described previously [31,32,33]. Rats were placed in an acrylic box placed on a 2 × 2 mm wire mesh test table and allowed to acclimate for at least 10 min. A monofilament was applied perpendicularly to the plantar surface of the affected paw for 3–5 s, which was performed 10 times for each monofilament. A response is considered positive if rats exhibits any behavior, such as licking or evasive reaction, more than 6 times. If it shows positive, it proceeds stimulating with the next smaller sized filament, and the smallest size that shows positive is set as a threshold. Each group was randomly mixed and only one experimenter performed the test, and the results were recorded by another experimenter.

### 4.8. Intrathecal Injection

Intrathecal injection was performed as follows [34]. Rats were placed in a prone position after anesthesia via Alfaxan. Then, 20 μL of solution (scrambled siRNA or PLGA-encapsulated CX3CR1 siRNA nanoparticles) was injected intrathecally through the intervertebral space between L5 and L6 using a Hamilton syringe (100 µL; Reno, NV, USA) mounted with a 26 gauge needle. Upon successful injection, obvious tail movement was observed. 

### 4.9. Statistical Analysis

The data are expressed as the mean ± standard error of the mean (SEM). Statistical significance in comparisons of multiple groups was assessed by *t*-test and one-way analysis of variance (ANOVA), followed by an appropriate multiple comparison test. *p*-values ≤ 0.05 were considered statistically significant. **p*-values < 0.05 and the data were analyzed by the Shapiro–Wilk test for normality. All statistical analyses were performed using GraphPad Prism 6 (GraphPad Software Inc., CA, USA).

## 5. Conclusions

Our data indicate that PLGA-encapsulated CX3CR1 siRNA nanoparticles effectively reduce neuropathic pain in SNL-induced rats by reducing microglial activity and expression of proinflammatory mediators. Therefore, we believe that PLGA-encapsulated CX3CR1 siRNA nanoparticles are valuable as a new treatment for neuropathic pain.

## Figures and Tables

**Figure 1 ijms-21-03469-f001:**
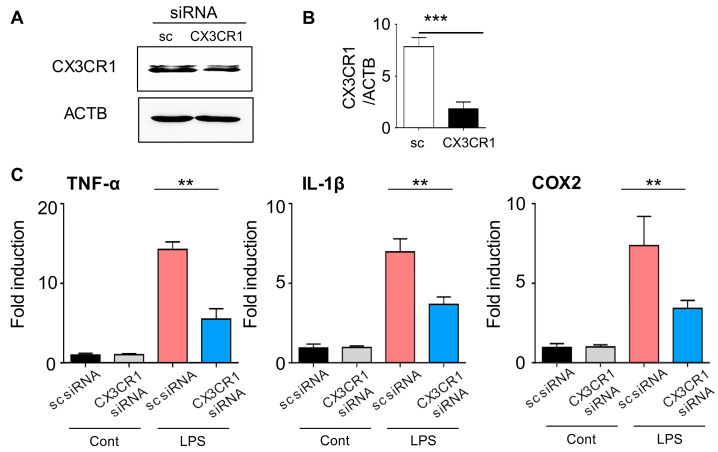
CX3CR1 siRNA inhibits CX3CR1 expression in BV2 cells and attenuates expression of proinflammatory mediators in lipopolysaccharide stimulated BV2 cells. (**A**,**B**) After transfection of BV2 cells with CX3CR1 siRNA or scrambled control siRNA (sc siRNA) for 2 days, the expression of CX3CR1 protein was detected by Western blot using an anti-CX3CR1 antibody, and the CX3CR1/ACTB ratio was quantified. Data are presented as the mean ± SEM (*t*-test ****p* < 0.001 versus sc siRNA). (**C**) sc siRNA or CX3CR1 siRNA was transfected into BV2 cells for 2 days. mRNA levels of TNF-α, IL-1β, and COX-2 were compared based on the presence or absence of LPS and quantified by qRT-PCR. Data are presented as the mean ± SEM (one-way ANOVA with Tukey’s post hoc test, ***p* < 0.01 versus sc siRNA + LPS). Cont, control; LPS, lipopolysaccharide.

**Figure 2 ijms-21-03469-f002:**
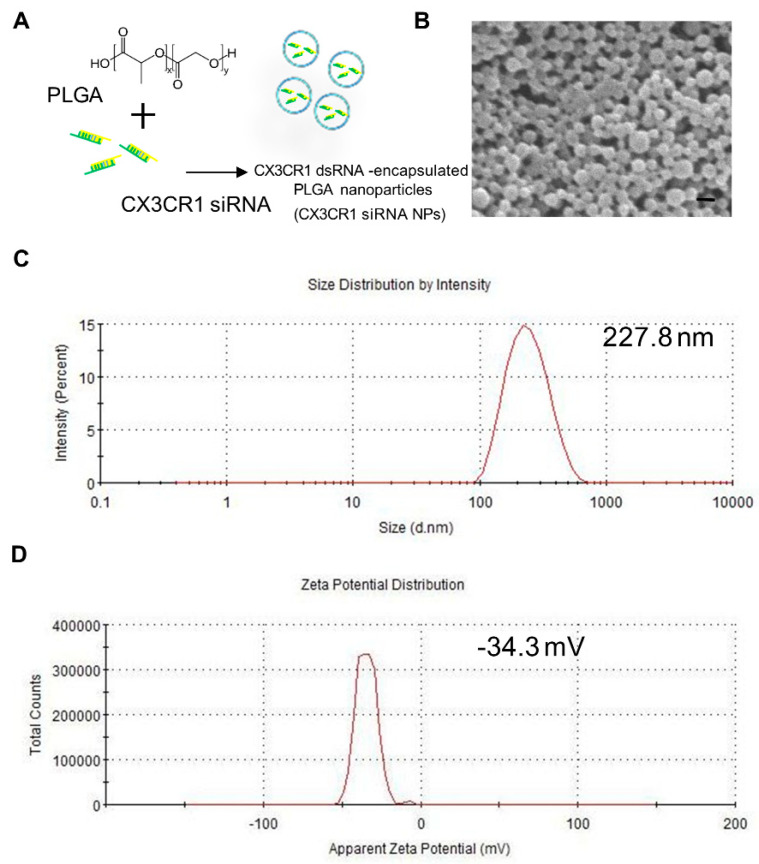
Characterization of siRNA-encapsulated Poly(D,L-lactic-co-glycolic acid) (PGLA) nanoparticles. (**A**) siRNA-encapsulated PLGA nanoparticles were prepared by sonicating a mixture of PGLA and CX3CR1 siRNA. (**B**) Nanoparticles were assessed by scanning electron microscope (SEM), and particle size (**C**) and zeta potential (**D**) were examined using a Zetasizer Nano ZS. Scale bar = 300 nm.

**Figure 3 ijms-21-03469-f003:**
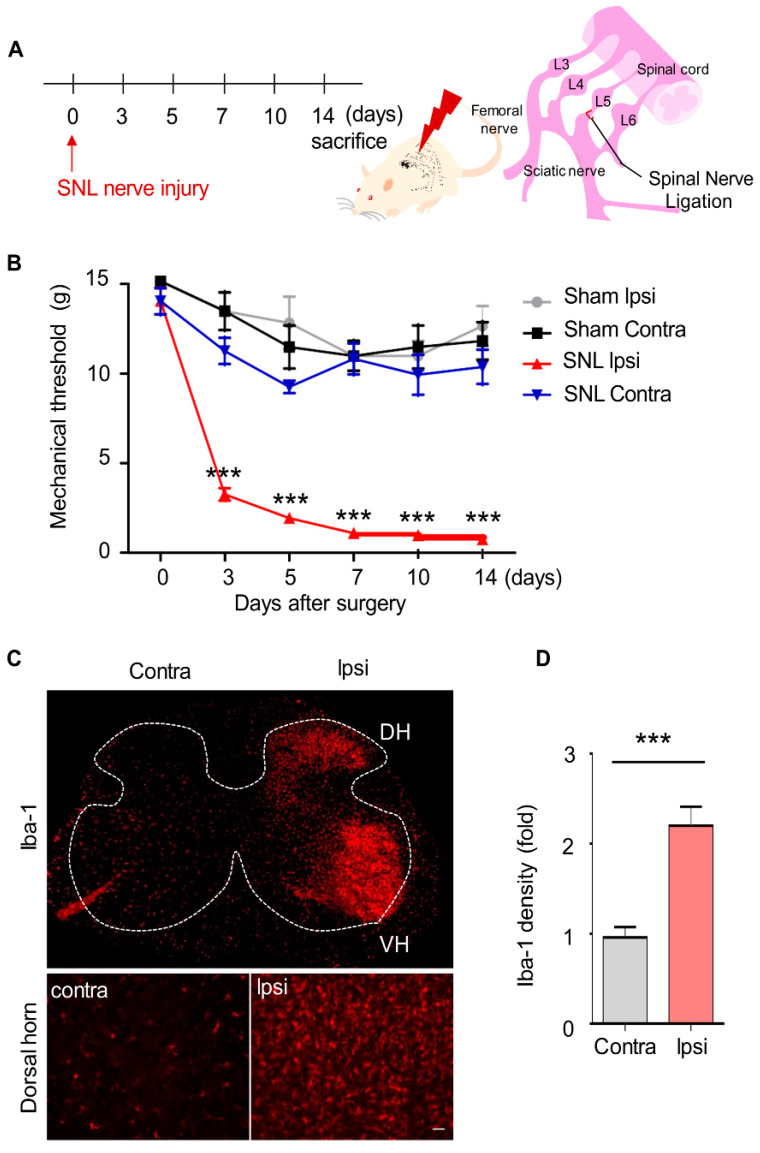
Mechanical allodynia and upregulated microglia activation in spinal nerve ligation-induced rats. (**A**) Neuropathic pain in rats was induced by spinal nerve ligation at the L5 vertebra. (**B**) Afterwards, the rats were subjected to a pain behavior test using von Frey filaments to evaluate the development of neuropathic pain. Data are presented as the mean ± SEM (one-way analysis of variance (ANOVA) with Dunnett’s post hoc test, ****p* < 0.001 versus Ipsi), *n* = 8 per group (**C**,**D**) At 3 days post-SNL surgery, L5 spinal sections were made and immuno-stained with anti-Iba-1 antibody (a microglia-specific marker). Scale bar = 150 μm (top), 100 μm (bottom). Data are presented as the mean ± SEM (*t*-test ****p* < 0.001 versus Contra), *n* = 8 per group SNL, spinal nerve ligation; Contra, contralateral; Ipsi, ipsilateral; DH, dorsal horn; VH, ventral horn.

**Figure 4 ijms-21-03469-f004:**
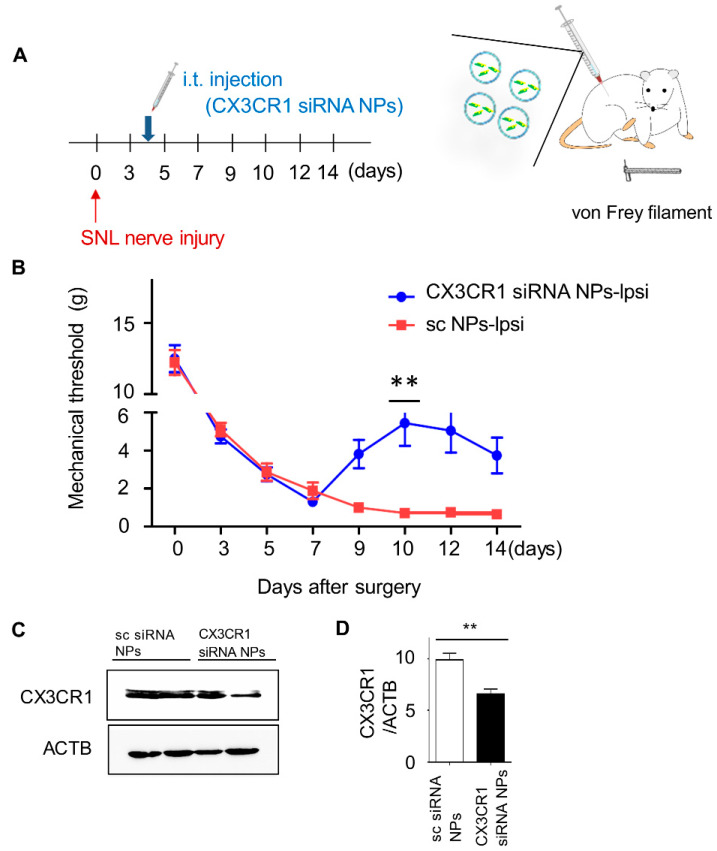
Intrathecal injection of PLGA-encapsulated CX3CR1 siRNA nanoparticles alleviates mechanical allodynia in spinal nerve ligation-induced rats. (**A**) sc siRNA or CX3CR1 siRNA nanoparticles were injected intrathecally at 4 days post-SNL nerve injury. (**B**) Rats were then subjected to pain behavior testing using von Frey filaments to evaluate the effect of CX3CR1 siRNA on neuropathic pain. Data are presented as the mean ± SEM (one-way ANOVA with Dunnett’s post hoc test, ***p* < 0.01 versus sc NPs-Ipsi), *n* = 7 per group. (**C**) At 6 days post-intrathecal injection, L5 spinal sections were made, and expression of CX3CR1 protein was detected by Western blot using an anti-CX3CR1 antibody, and the CX3CR1/ACTB ratio was quantified. (**D**) Data are presented as the mean ± SEM (*t*-test, one-way ANOVA, ***p* < 0.01 versus sc siRNA NPs), *n* = 7 per group NPs, nanoparticles; i.t, intrathecal; sc, scrambled; Ipsi, ipsilateral.

**Figure 5 ijms-21-03469-f005:**
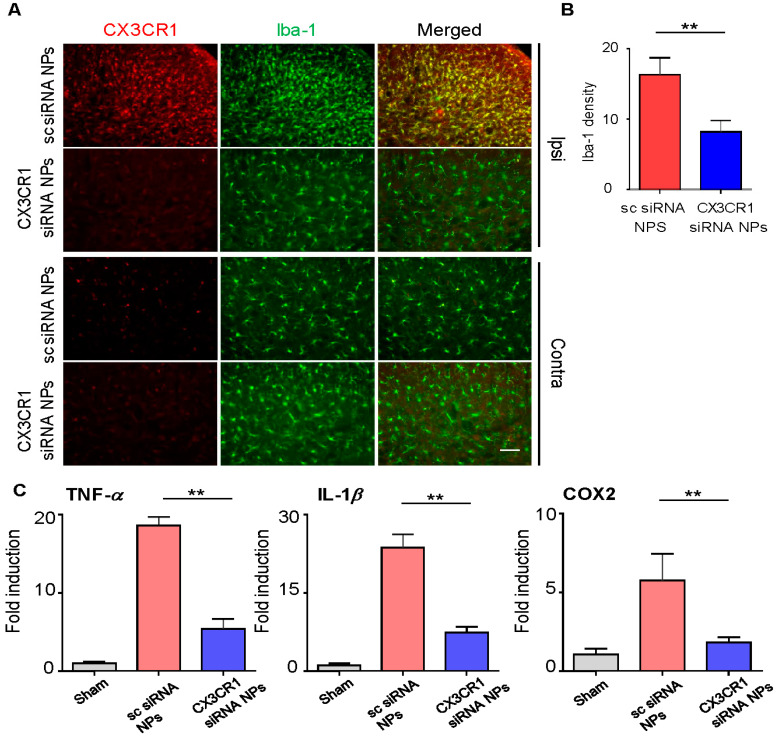
PLGA-encapsulated CX3CR1 siRNA nanoparticles attenuate expression of proinflammatory mediators in SNL rats. (**A**,**B**) At 6 days post-intrathecal injection, L5 spinal sections were made and incubated with anti-Iba-1 and anti-CX3CR1 antibodies. Data are presented as the mean ± SEM (*t*-test, ***p* < 0.01 versus sc siRNA NPs), *n* = 7 per group. Scale bar = 50 μm (**C**) On day 6 post-injection, total mRNA was extracted from the ipsilateral spinal dorsal horn at L4–5 (0.7 cm) and utilized for cDNA synthesis. mRNA levels of TNF-α, IL-1β, and cyclooxygenase (COX) 2 were then measured using qRT-PCR. Data are presented as mean ± SEM (one-way ANOVA with Tukey’s post hoc test, ***p* < 0.01 versus sc siRNA NPs), *n* = 7 per group. sc, scrambled; NPs, nanoparticles.

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
