# Peer review of "CX3CR1-Targeted PLGA Nanoparticles Reduce Microglia Activation and Pain Behavior in Rats with Spinal Nerve Ligation"

_ijms, 2020, doi:10.3390/ijms21103469_

Round 1

Reviewer 1 Report

In the present study the Authors have evaluated the effect of CX3CR1 siRNA in SNL model in rats.  CX3CR1 siRNA-treatment reduced mechanical allodynia together with reduction of spinal microglia activation and downregulation of some proinflammatory mediators. The topic is quite interesting; however I have several issues that need to be addressed.

After SNL surgery, microglia seem to be activated also in the ipsilateral ventral horn of the spinal cord. Please clarify. Have the Authors performed morpho-functional analysis of microglia?

Have the Authors evaluated CXCL3 expression?

Have the Authors evaluated possible indirect effects of CX3CR1 siRNA on astrocytes morpho-functioning?

Results section must be improved. It should be revised by remodelling the order of the sections in a logical sequence. Revise the title of section 2.1.

A statistical analysis section including N of animals or samples, specific tests used (Anova or T-test…) and P and F values should be included. Please remove N in the graphs.

Discussion should be entirely revised. Data reported in the study have been not accurately discussed.

Author Response

We appreciate your careful review and comments on our manuscript. We have carefully revised our manuscript according to your suggestions. (Revised parts are marked in red in the manuscript.) 

Reviewer 2 Report

In this study, the use of PLGA-encapsulated CX3CR1 siRNA nanoparticles for the treatment of neuropathic pain in a rat model of SNL has been investigated. Authors have written a clear introduction, designed a rational study and results are promising. The discussion is fair and methods are described clearly. There are number of points to consider for revisions to enhance the clarity. Please see below:

·       Please add a reason for only including rats that responded to von frey application of 10g. Line 92, page 2.

·       It seems that activation of microglia in contralateral side is time dependent, Line 97, page 3. Is that correct?

·       protein was reduced by ~50% in treated condition, line 119, page 4. What is the definition of efficacy here? Is that defined a minimum requirement? An optimal reduction in the literature or is that model-dependent?

·       In vitro tests of LPS-treated BV2 cells also showed the effect of the treatment. Can authors explain why LPS, as that is an inflammatory type substance rather an injury. Does this mean that authors are trying to emphasize that the treatment can be used for both inflammatory and neuropathic type of pain? Please expand the information on in vitro part.

·       Injection of encapsulated has been done 3 days post-surgery, is that a specific reason for that? Is that because peak of effect of the surgery is seen 3 days after? Please add if so, around like 166 page 6. What about prophylaxis injection, i.e. injection the substance before surgery and compared with after? What the authors assume about the outcome?  

·       L5 spinal dorsal horn tissues have been collected 6 days post-intrathecal injection and results show a reduction in the expression of CX3CR1 protein (line 174, page 6). Authors have continued until day 14 after injection for observing the behavioral effects. Did they also obtained samples from the spinal cord for WB at day 14? The question is why day 6 as a time point to check.

·       Did the author look at the contralateral part as well?

·       Only male animals have been used in the study. While recent recommendation is the application of both sexes after identification of differences in glial cell activation and pathways in pain in female and male rats. Can author elaborate further in choice of animals from one sex here? Line 253, page 9.

·       Please add the total number of animals used. It is also suggested that for further clarification, it is a good practice to add the N to all figs or fig legends, when appropriate. Sometimes, it is assumed that if groups of rats are used with 6, would always be 6, however, in some experiments, N might be 5, because of a loss or other methodological complications that they do not allow for taking some data or samples from number of rats within the group of 6, hence leaving the final actual tests on lower number of animals per group. This needs transparency in reporting.

·       Please briefly add the procedure for von Frey test in the method section (line 316, page 10). It is right now just by refereeing the readers to previous publications. It is at least important to add whether it was manual or electronic von Frey and was the procedure done blinded for the investigator testing sham, and treatment groups? Randomization and blinding of animals are recommended in ARRIVE guidelines. Authors are encouraged to add those information, if they have properly followed the ARRIVE guidelines.

·       CX3CR1 siRNA-encapsulated PLGA nanoparticles were injected intrathecal (line 318, page 10). Can authors explain if any complications or side effects were noticeable?

·       For statistical analysis (section 4.9, page 11), it is reflected that data were tested for normality for example by Shapiro-Wilk test? It is assumed because authors presented data with mean and SD and used parametric tests, like t test and ANOVA. Add the normality test if you did, please. If data were not normal, median and IQ must be reported and non-parametric, tests must be used, like Friedman instead of ANOVA. If authors have had a data set of non-normal distribution and log transformed data to get to normal and use parametric statistics, it must also be stated. Earlier in result, it is read that t test was used, but in the statistic section, it is not indicated. Please add all tests that you used in the result section.

·       At the end of discussion, methodological considerations, limitations of this study and ways for translation of data into clinic can be added. Authors are encouraged to highlight the novelty and value of their findings and the potential as how this can be used in the clinic for patients, and what challenges will be faced in future study. In this way, a future perspective to this stud can be outlined for encouraging continuation in the field.

Author Response

We appreciate your careful review and comments on our manuscript. We have carefully revised our manuscript according to your suggestions. (Revised parts are marked in red in the manuscript.) 

Response to Reviewer 2 Comments

Manuscript # ijms-797012

TITLE: “CX3CR1-targeted PLGA nanoparticles reduce microglia activation and pain behavior in rats with spinal nerve ligation” by Noh et al.

Dear Reviewer 2,

We appreciate your careful review and comments on our manuscript. We have carefully revised our manuscript according to your suggestions. (Revised parts are marked in red in the manuscript.) 

Comments to the Author:

 Please add a reason for only including rats that responded to von frey application of ≥10g. Line 92, page 2.

  • According to your suggestion, we added content to manuscript.

[1]

It seems that activation of microglia in contralateral side is time dependent, Line 97, page 3. Is that correct?

  • Thanks for your comment, we have revised the manuscript as suggested in order to avoid concerns.

protein was reduced by ~50% in treated condition, line 119, page 4. What is the definition of efficacy here? Is that defined a minimum requirement? An optimal reduction in the literature or is that model-dependent?

  • We are sorry for this mistake. The above sentence is an error that occurred during the English proofreading process. Therefore, it was deleted from the manuscript.

In vitro tests of LPS-treated BV2 cells also showed the effect of the treatment. Can authors explain why LPS, as that is an inflammatory type substance rather an injury. Does this mean that authors are trying to emphasize that the treatment can be used for both inflammatory and neuropathic type of pain? Please expand the information on in vitro part.

  • Thanks for your question. The TLR4 / NF-kB pathway is activated by LPS[2] and the CX3CL1-CX3CR1 axis is linked to the Nf-kB pathway[3]. Accordingly, LPS stimulation induces the expression of various pro-inflammatory genes including cytokines and chemokines, and participates in inflammation control. Therefore, we attempted to confirm the anti-inflammatory effect of CX3CR1 siRNA through LPS.

Injection of encapsulated has been done 3 days post-surgery, is that a specific reason for that? Is that because peak of effect of the surgery is seen 3 days after? Please add if so, around like 166 page 6. What about prophylaxis injection, i.e. injection the substance before surgery and compared with after? What the authors assume about the outcome?  

  • According to your suggestion, we added content to manuscript.

  • Since neuropathic pain occurs after injury, prophylactic drug infusion was not considered because it would be considered clinically impossible.

L5 spinal dorsal horn tissues have been collected 6 days post-intrathecal injection and results show a reduction in the expression of CX3CR1 protein (line 174, page 6). Authors have continued until day 14 after injection for observing the behavioral effects. Did they also obtained samples from the spinal cord for WB at day 14? The question is why day 6 as a time point to check.

  • Thanks for your reminder. This is because in this study, the pain relief effect was the best on the 6th day after injection. we added content to manuscript.

Did the author look at the contralateral part as well?

  • Thanks for your question. According to your suggestions added to the figure.

Only male animals have been used in the study. While recent recommendation is the application of both sexes after identification of differences in glial cell activation and pathways in pain in female and male rats. Can author elaborate further in choice of animals from one sex here? Line 253, page 9.

  • Thank you for your interesting question. Since female rat behavior can be influenced by the estrous cycle, only male rats were used to eliminate possible variables in experimental conditions.[4]

Please add the total number of animals used. It is also suggested that for further clarification, it is a good practice to add the N to all figs or fig legends, when appropriate. Sometimes, it is assumed that if groups of rats are used with 6, would always be 6, however, in some experiments, N might be 5, because of a loss or other methodological complications that they do not allow for taking some data or samples from number of rats within the group of 6, hence leaving the final actual tests on lower number of animals per group. This needs transparency in reporting.

  • Thank you for your comment, we have revised the “Materials and methods” section in the manuscript as suggested.

Please briefly add the procedure for von Frey test in the method section (line 316, page 10). It is right now just by refereeing the readers to previous publications. It is at least important to add whether it was manual or electronic von Frey and was the procedure done blinded for the investigator testing sham, and treatment groups? Randomization and blinding of animals are recommended in ARRIVE guidelines. Authors are encouraged to add those information, if they have properly followed the ARRIVE guidelines.

  • According to your suggestion, we added content to manuscript.

CX3CR1 siRNA-encapsulated PLGA nanoparticles were injected intrathecal (line 318, page 10). Can authors explain if any complications or side effects were noticeable?

  • Thank you for your interesting question. Since there is no way to find a noticeable side effect in intrathecal injection performed in the rat model, the successful injection was evaluated through tail movement.

For statistical analysis (section 4.9, page 11), it is reflected that data were tested for normality for example by Shapiro-Wilk test? It is assumed because authors presented data with mean and SD and used parametric tests, like t test and ANOVA. Add the normality test if you did, please. If data were not normal, median and IQ must be reported and non-parametric, tests must be used, like Friedman instead of ANOVA. If authors have had a data set of non-normal distribution and log transformed data to get to normal and use parametric statistics, it must also be stated. Earlier in result, it is read that t test was used, but in the statistic section, it is not indicated. Please add all tests that you used in the result section.

  • Thanks for your kind comment. We corrected the description of statistical analysis. We have addressed the normality test in the revised manuscript. Actually, we did the Shapiro-wilk test as normality test using Graphpad Prism 6.

At the end of discussion, methodological considerations, limitations of this study and ways for translation of data into clinic can be added. Authors are encouraged to highlight the novelty and value of their findings and the potential as how this can be used in the clinic for patients, and what challenges will be faced in future study. In this way, a future perspective to this stud can be outlined for encouraging continuation in the field.

  • Thank you for your comments. we have revised in the manuscripts and added

References

  1. Shin, J.; Yin, Y.; Park, H.; Park, S.; Triantafillu, U.L.; Kim, Y.; Kim, S.R.; Lee, S.Y.; Kim, D.K.; Hong, J.J.N. p38 siRNA-encapsulated PLGA nanoparticles alleviate neuropathic pain behavior in rats by inhibiting microglia activation. 2018, 13, 1607-1621.
  2. Liu, T.; Zhang, L.; Joo, D.; Sun, S.-C.J.S.t.; therapy, t. NF-κB signaling in inflammation. 2017, 2, 1-9.
  3. Ding, X.-M.; Pan, L.; Wang, Y.; Xu, Q.-Z.J.I.j.o.m.m. Baicalin exerts protective effects against lipopolysaccharide-induced acute lung injury by regulating the crosstalk between the CX3CL1-CX3CR1 axis and NF-κB pathway in CX3CL1-knockout mice. 2016, 37, 703-715.
  4. Chung, J.M.; Kim, H.K.; Chung, K. Segmental spinal nerve ligation model of neuropathic pain. In Pain Research, Springer: 2004; pp. 35-45.

Round 2

Reviewer 1 Report

The manuscript has been improved. Please, insert the discussion section before conclusions. 

Reviewer 2 Report

The authors have revised the manuscript carefully based on the raised comments. There is no more comment.